# Relationship between Composite Dietary Antioxidant Index and Aging

**DOI:** 10.3390/healthcare11202722

**Published:** 2023-10-12

**Authors:** Haiting Wang, Yongbing Chen

**Affiliations:** 1Beijing International Science and Technology Cooperation Base for Cancer Metabolism and Nutrition, Beijing 100038, China; 18612583366@163.com; 2Department of Gastrointestinal Surgery, Beijing Shijitan Hospital, Capital Medical University, Beijing 100054, China; 3Department of Clinical Nutrition, Beijing Shijitan Hospital, Capital Medical University, Beijing 100054, China

**Keywords:** dietary antioxidant index, aging, phenotypic age, NHANES

## Abstract

Background: Numerous studies have demonstrated a close relationship between antioxidant-rich diets and comorbidities as well as mortality. However, the relationship between such diets and aging remains unclear. The purpose of this study was to investigate the association between the Composite Dietary Antioxidant Index (CDAI) and aging. Methods: All participants were from the National Health and Nutrition Examination Survey (NHANES) 2001–2010. Phenotypic age was calculated using a formula and subtracted from the chronological age to determine the aging. When the phenotypic age exceeded the chronological age, it was considered as aging. A weighted logistic regression model was employed to explore the relationship between CDAI and aging. Restricted cubic splines (RCSs) were used to examine the potential nonlinear relationship between them. Subgroup analysis and joint analysis were conducted to explore the effect of modifiers in these relationships. Results: A total of 19,212 participants (weighted: 165,285,442 individuals) were included in this study. The weighted logistic regression model showed a significant correlation between CDAI and the risk of aging (OR = 0.90, 95% CI: 0.84–0.96). RCS analysis revealed an L-shaped dose–response relationship between CDAI and the risk of aging. Subgroup analysis indicated that the association between CDAI and aging was more pronounced in middle-aged individuals and non-smokers. The joint analysis demonstrated that although smoking accelerated aging among participants, a high CDAI diet could still offset these damages. Conclusions: The association between high CDAI and reduced risk of aging is particularly significant in young and middle-aged individuals and non-smokers. Consuming foods rich in CDAI components may potentially lower the risk of aging.

## 1. Introduction

With the development of society and the improvement in medical standards, human life expectancy is continuously increasing. According to the latest report by the World Health Organization (WHO), by 2030 approximately one-sixth of the global population may reach the age of 60 or above. Furthermore, by 2050 this population is projected to double [1]. Generally, aging is often accompanied by a decline in physiological function and an increase in chronic diseases. Research suggests that 23% of the global disease burden can be attributed to diseases among individuals aged 60 and above. Moreover, as age increases, the burden of disease progressively intensifies [2]. Therefore, the issue of population aging poses a significant challenge to public health security. However, aging is not merely the result of increased chronological age and declining organ function; it also involves genetic alterations and disruptions in protein functionality at the molecular level, as well as metabolic dysregulation and an increased prevalence of chronic inflammation [3]. Due to the complexity of the aging process, it requires a multifaceted approach for assessment and evaluation. Merely relying on the increase in chronological age seems insufficient to accurately determine physiological aging within the body [4]. In recent years, a novel concept called “PhenoAge” has been introduced. It is a measurement method that incorporates multiple indicators to calculate an individual’s age, providing a better representation of their actual age and physical condition [5,6]. Multiple studies have validated the relationship between PhenoAge and comorbidity and mortality [6,7].

Biologists aim to intervene, slow down, or even reverse the aging process through various measures. It has been proven that environmental interventions have a significant impact on aging. In particular, dietary modifications have shown to be particularly effective in combating aging [8]. We have observed that supplementation with individual nutrients may be closely associated with anti-aging effects. Eugenio et al. demonstrated that supplementation of vitamin E can effectively counteract aging and the inflammatory burden associated with aging [9]. We acknowledge that while individual nutrient supplementation can be useful, people’s diets are holistic and require proper combinations and compositions. It is necessary to explore effective dietary patterns and scoring systems for anti-aging purposes. The Composite Dietary Antioxidant Index (CDAI) is widely used to assess the antioxidant capacity of dietary components. It is calculated based on the levels of vitamin A, C, and E, as well as manganese, selenium, and zinc [10]. However, there has been limited exploration of the relationship between CDAI and aging. Therefore, the aim of this study is to investigate the potential association between CDAI and aging using the National Health and Nutrition Examination Survey (NHANES) database. This study aims to provide robust dietary guidance for anti-aging purposes.

## 2. Materials and Methods

### 2.1. Study Participants

All patients in this study were sourced from NHANES, an ongoing research initiative conducted by the National Center for Health Statistics aimed at investigating the health and nutrition of the U.S. population. NHANES employs a multi-stage, complex probability sampling methodology to collect representative health data for the U.S. population, as described earlier [11]. For this cross-sectional study, we included individuals from five survey cycles conducted between 2001 and 2010 who had complete dietary information. Exclusion criteria included the following. (1) Participants lacking covariates required for the calculation of phenotypic age (albumin, creatinine, glucose, C-reactive protein [CRP], lymphocyte percentage, mean cell volume, red cell distribution width, alkaline phosphatase, and white blood cell count). (2) Participants lacking covariates for model adjustments (factors such as smoking, alcohol consumption, comorbidity information, health insurance, and income, Appendix A). After applying the strict inclusion and exclusion criteria, a total of 19,212 participants were included in this study (Appendix A).

### 2.2. Exposures and Covariates

Dietary data were obtained through 24 h dietary recall interviews conducted with participants. They were asked detailed questions about their diet at the mobile examination center (MEC), where trained staff recorded the participants’ dietary information. To ensure the reliability of the data source, all participants were asked to provide a second report via telephone 3–10 days later. The mean values of both reports were included in this study. The calculation of CDAI was performed using a previously validated formula. This index incorporates six minerals or vitamins associated with oxidative processes [12]. The formula is as follows:$$\mathrm{CDAI}=\sum_{i=1}^{n=6}\left(Total\;nutrient\;intakes-Mean\right)/SD$$

Total nutrient intake represents the average values of each individual’s CDAI components from the two visits. Mean represents the average values of each CDAI component across the entire cohort, while *SD* represents the standard deviation of each CDAI component across the entire cohort.

The covariates included in the study are age, gender, race, waist circumference, education level, poverty income ratio (PIR), health insurance, smoking, alcohol consumption, diabetes mellitus (DM), cardiovascular disease (CVD), hypertension, history of cancer, CRP, and albumin. The race categories include Non-Hispanic Black, Non-Hispanic White, and other races (Mexican American, Other Hispanic, or Other Race—Including Multi-Racial). Diabetes was defined as fasting blood glucose levels ≥ 7 mmol/L or the use of oral hypoglycemic medications.

### 2.3. Outcome

Morgan E. Levine et al. utilized the NHANES III to calculate phenotypic age and validated it using the NHANES IV [5]. Phenotypic age is determined by computing nine hematological parameters (albumin, creatinine, glucose, [log] CRP, lymphocyte percentage, mean cell volume, red cell distribution width, alkaline phosphatase, and white blood cell count) along with chronological age. The formula for calculating phenotypic age is as follows:(1)xb = −19.907 − 0.0336 × albumin − 0.0095 × creatinine − 0.0195 × glucose + 0.0954 × ln (CRP) − 0.0120 × lymphocyte percent + 0.0268 × mean cell volume + 0.3356 × red blood cell distribution width + 0.00188 × alkaline phosphatase + 0.0554 × white blood cell count + 0.0804 × chronological age.(2)Phenotypic Age = 141.50 +ln [−0.0053 × ln (1 − xb)]/0.09165.

Finally, in order to differentiate between individual’s chronological age and phenotypic age difference, Zuyun Liu et al. also calculated a metric called phenotypic age acceleration: PhenoAgeAccel = Phenotypic Age − Chronological Age. We define phenotypic aging as when PhenoAgeAccel is greater than 0, and phenotypic rejuvenation as when it is less than or equal to 0 [6].

### 2.4. Statistic Analysis

Since NHANES is a stratified and multi-stage sampling survey, we applied appropriate sampling weights to each participant during the statistical analysis to ensure the representativeness of the population. Descriptive statistics were used to summarize the data, where normally distributed variables were presented as mean ± standard deviation, and skewed variables were expressed as median (interquartile range) or frequency (percentage). In cases of skewed distribution, the values were transformed using the natural logarithm. To examine the linear relationship between CDAI and phenotypic age, Pearson correlation analysis was conducted. Furthermore, the relationship between CDAI or its components and aging was assessed using restrictive cubic spline regression (RCS). Weighted logistic regression analysis was employed to describe the association between CDAI or its components and aging, presented as odds ratio (OR) with corresponding 95% confidence intervals (CI). In the logistic regression analysis, we examined the relationship between CDAI as a continuous variable, as a binary variable (based on the median), and as a categorical variable with five groups (based on quintiles) and aging. We constructed three models to comprehensively evaluate the relationship between them. Model 1 was a crude model. Model 2 was adjusted for sex, race, BMI, education level, PIR, health insurance, smoke, and alcohol. Model 3 was adjusted for model 2 and DM, CVD, hypertension, cancer history, CRP, and albumin.

Interaction analysis was employed to explore the potential modifiers in these relationships, including age, sex, race, PIR, health insurance, smoking, alcohol consumption, DM, hypertension, cancer, and CRP (with a cutoff of 1 mg/L). Subgroup analyses were conducted based on these factors to describe the relationship between CDAI and aging across different strata. Furthermore, joint analyses were performed to further elucidate the role of the modifiers in the presence of interaction effects.

Sensitivity analyses were conducted to assess the robustness of the findings. These included excluding participants with a history of cancer (as tumor burden may significantly impact nutritional status and levels of inflammation, thereby affecting phenotypic age [13]), excluding participants older than 75 years, and including data from three examinations conducted between 2001 and 2006. The statistical analysis was conducted using R 4.2.0 software, and statistical significance was determined with a two-sided *p*-value of less than 0.05.

## 3. Results

### 3.1. Baseline Characteristics

A total of 19,212 participants (weighted: 165,285,442 individuals) were included in this study (Table 1). The average age was 46.49 ± 16.2 years, with 49.7% (9821 individuals) being male. The average level of CDAI was 0.52. Participants were divided into five groups (Q1 to Q5) based on quintiles of CDAI: Q1 (−7.35 to −2.63), Q2 (−2.63 to −1.01), Q3 (−1.01 to 0.65), Q4 (0.65 to 3.06), and Q5 (3.06 to 36.70).

Compared to participants in Q1, those in Q5 had a lower phenotypic age; a higher proportion of males; smaller waist circumference; higher education levels; higher income levels; a lower prevalence of smoking; fewer cases of cardiovascular disease (CVD), DM, and hypertension; lower inflammation levels; and higher levels of albumin.

### 3.2. Association of Composite Dietary Antioxidant Index and Aging

We initially investigated the correlation between phenotypic age and CDAI (Appendix A). The results showed a negative correlation between CDAI and phenotypic age across different age groups, genders, BMI categories, and races. The results from the restrictive cubic spline regression (RCS) analysis also demonstrated an L-shaped dose–response relationship between CDAI and the risk of aging (Figure 1).

Table 2 presents the results of the multivariable logistic analysis assessing the association between CDAI and the risk of aging. When treated as a continuous variable and adjusted for confounding factors (sex, race, BMI, education level, PIR, health insurance, smoking, alcohol, DM, CVD, hypertension, cancer history, CRP, and albumin), each standard deviation increase in CDAI was associated with an OR of 0.90 (95% CI: 0.84–0.96) for aging. Even after categorizing CDAI using cutoff values and quintiles, a negative correlation between CDAI and aging persisted. Compared to the low CDAI group, the high CDAI group had a lower risk of aging (OR: 0.82, 95% CI: 0.72–0.93). Additionally, there was a decreasing trend in aging risk across the different quintiles: Q1 (reference), Q2 (OR: 0.89, 95% CI: 0.76–1.04), Q3 (OR: 0.87, 95% CI: 0.70–1.06), Q4 (OR: 0.82, 95% CI: 0.69–0.98), and Q5 (OR: 0.68, 95% CI: 0.57–0.81). The trend test showed statistically significant differences (*p* = 0.014).

In addition, to further elucidate the relationship between CDAI and aging, we also examined the associations between CDAI components and the risk of aging. We found that, unlike CDAI as a whole, only vitamin C, vitamin E, selenium, and manganese showed an L-shaped relationship with the risk of aging, while vitamin A and zinc did not demonstrate such a pattern (Appendix A). The results of COX regression analysis further supported these findings. Vitamin C (OR: 0.94, 95% CI: 0.89–0.99), vitamin E (OR: 0.86, 95% CI: 0.80–0.92), selenium (OR: 0.89, 95% CI: 0.83–0.96), and manganese (OR: 0.89, 95% CI: 0.84–0.94) were significantly associated with a decreased risk of aging (Appendix A).

### 3.3. Subgroup Analysis and Joint Analysis

We conducted a detailed subgroup analysis based on age and found an interesting observation. The relationship between CDAI and the risk of aging was significant in the middle-aged and younger population, rather than in individuals aged over 60 years (Table 3). When considering CDAI as a continuous variable, the odds ratios (ORs) for the risk of aging associated with CDAI were 0.89 (95% CI: 0.80–0.99) and 0.90 (95% CI: 0.80–1.00) in participants aged <45 years and those aged 45–59 years, respectively. However, in participants aged 60–74 years and 75–84 years, the ORs for the risk of aging associated with CDAI were 0.91 (95% CI: 0.80–1.05) and 0.95 (95% CI: 0.8–1.10), respectively. The results when considering CDAI as quintiles were similar. Compared to participants in Q1, in the Q5 group of participants aged <45 years and those aged 45–59 years, the odds ratios (ORs) for the risk of aging associated with CDAI were 0.65 (95% CI: 0.48–0.89) and 0.69 (95% CI: 0.48–0.98), respectively. However, in the Q5 group of participants aged 75–84 years, the OR for the risk of aging associated with CDAI was 0.85 (95% CI: 0.52–1.38). Furthermore, we performed subgroup and interaction analyses based on gender, race, PIR, health insurance, smoking, alcohol consumption, diabetes, hypertension, cancer, and CRP (Figure 2). The results indicated an interaction between smoking and the relationship between CDAI and the risk of aging (*p* = 0.035). In non-smokers, the negative association between CDAI and the risk of aging was more pronounced. Once the interaction effect was determined, we conducted joint analyses (Figure 3). The results demonstrated that although smoking accelerates aging and increases the risk of aging, a high CDAI can partially counteract these negative effects.

### 3.4. Sensitivity Analyses

To assess the robustness of the results, sensitivity analyses were conducted. After excluding participants with a history of cancer, the association between CDAI and the risk of aging remained significant (OR: 0.91, 95% CI: 0.85–0.96). Similarly, when individuals aged over 75 years were excluded, the results remained robust (OR: 0.89, 95% CI: 0.84–0.96). Furthermore, an internal validation was performed using participants from the years 2001–2006 as a validation set. Although the odds ratio for CDAI as a continuous variable was at the margin (OR: 0.92, 95% CI: 0.83–1), the trend of a decreased risk of aging with higher CDAI values persisted when considering CDAI as a categorical variable (Appendix A).

## 4. Discussion

In this large cross-sectional study using data from NHANES between 2001 and 2010, we found that higher levels of CDAI were associated with a decreased risk of aging. Importantly, this relationship was more pronounced in the middle-aged and younger population. Subgroup and joint analyses revealed that the association between CDAI and the risk of aging was influenced by smoking. However, overall, despite the detrimental effects of smoking on aging, CDAI was able to partially counteract these effects. Overall, our findings suggest that CDAI may serve as a potential indicator of anti-aging effects. Additionally, our study underscores the importance of considering lifestyle factors, such as smoking, in understanding the complex relationship between CDAI and the risk of aging.

Several studies have highlighted the importance of CDAI in various aspects. Wang et al. demonstrated that higher CDAI levels were associated with a reduced risk of all-cause mortality and cardiovascular mortality. They suggested that a dietary intake rich in antioxidants may significantly lower the mortality rate from cardiovascular diseases [12]. A study from Italy has indicated that a high intake of antioxidant-rich diet may reduce the risk of HPV infection and HPV-related diseases [14]. Regarding cancer, Yu et al. demonstrated that higher levels of CDAI were associated with a reduced risk of colorectal cancer [15]. Consistent with our study, they also found that this inverse association was primarily observed in individuals who never smoked and showed a significant interaction effect. However, to our knowledge, our study is the first specifically to investigate the relationship between CDAI and aging.

The relationship between oxidative stress and aging has been extensively studied in both clinical and basic research, as oxidative stress is considered a significant factor in the aging process [16]. Oxidative stress refers to a state of cellular oxidative damage caused by the accumulation of excessive reactive oxygen species in both intracellular and extracellular environments. Inside of cells, sources of oxidative stress include the mitochondrial respiratory chain, cytochrome P450 enzyme system, and nucleic acid metabolism [17,18]. Outside of cells, oxidative stress can be induced by various factors such as inflammation, external environmental factors, and free radicals generated by metabolic activities within the organism [19]. Oxidative stress triggers a series of molecular mechanisms that can profoundly affect cellular function and viability. Among these mechanisms, oxidative damage to DNA, proteins, and lipids plays a crucial role in the aging process. DNA oxidative damage can lead to gene mutations and chromosomal instability, thereby impairing the cell’s ability to repair and regenerate itself [20]. Oxidative damage to proteins can result in functional dysregulation and abnormal aggregation, thereby affecting cellular metabolism and signaling pathways [21]. Additionally, the accumulation of lipid peroxidation products can cause damage to cellular membranes and functional abnormalities [22]. This intricate web of oxidative stress and its multifaceted impact on cellular components underscores its pivotal role in the aging process.

Oxidative stress exerts its influence not only by directly damaging cellular components but also by triggering a series of intricate signaling pathways that impact the cellular aging process. One such pathway that has been extensively studied is the nuclear factor-kappa B (NF-κB) pathway. When cells encounter oxidative stress, NF-κB is activated and subsequently translocates into the cell nucleus. In this nuclear location, NF-κB assumes its role as a regulator of gene expression, particularly for genes associated with inflammation and apoptosis. This regulation has significant consequences for cell survival and the initiation of inflammatory responses [23]. Another pivotal pathway linking oxidative stress and aging is the mitochondrial signaling pathway. Mitochondria, while essential for cellular energy production, are both a primary source of oxidative stress and highly susceptible to oxidative damage themselves. Any impairment in mitochondrial function can exacerbate oxidative stress within the cell, setting in motion a chain reaction that accelerates the aging process [24]. In summary, the relationship between oxidative stress and aging is a multifaceted and tightly interwoven one. Not only does oxidative stress directly damage cellular components, but it also sets off intricate signaling cascades involving pathways like NF-κB and mitochondria. Given the complexity of these interactions, the quest for potential interventions and the development of therapeutic strategies to counteract the aging process are of paramount importance. Such endeavors hold promise for extending healthy lifespans and enhancing overall well-being.

Diet plays a significant role in the anti-aging process, and its antioxidant-rich properties are crucial for mitigating oxidative stress damage and promoting healthy aging [25]. Antioxidants are bioactive compounds found in food that have the ability to neutralize free radicals and reduce oxidative damage [26]. Vitamin C, vitamin E, selenium, and other antioxidants are commonly recognized as key constituents of CDAI. Taking vitamin C as an example, research has shown that vitamin C is involved in the anti-aging process through multiple mechanisms. Importantly, it has the ability to neutralize free radicals and reduce oxidative stress-induced damage to cells. By scavenging and neutralizing free radicals, vitamin C helps maintain the cellular redox balance and mitigate the detrimental effects of oxidative stress on cells [27]. In addition, selenium also plays a role in antioxidant stress and anti-aging. It is a component of various enzymes that participate in regulating the redox balance and combating oxidative stress within cells. These enzymes include glutathione peroxidase (GPx), selenoprotein P (SelP), and selenoprotein W (SelW), among others. Selenium acts as a cofactor for these enzymes, enhancing their antioxidant capacity and contributing to the overall protection against oxidative damage and aging [28,29]. GPx, as an crucial antioxidant enzyme, plays a essential role in cellular defense against oxidative stress. It catalyzes the reaction between substrates such as glutathione (GSH) and hydrogen peroxide, converting harmful peroxides into harmless water and alcohol, thus reducing oxidative stress within cells. By reducing free radical levels, GPx helps protect the structure and function of cells from oxidative damage and contributes to delaying the aging process [30].

This study also has the following limitations. Firstly, as with all cross-sectional studies, we only observed the relationship between CDAI and the risk of aging, and causality cannot be determined. Prospective studies are needed to establish causal relationships. Secondly, we did not account for the use of antioxidants, which is a general term for drugs with antioxidant properties [31] This could potentially lead to an underestimation of the risk of aging and the ability of CDAI. Importantly, we only included dietary data from a single time point, which may introduce bias. Long-term dietary investigations are necessary to provide more accurate assessments.

## 5. Conclusions

The association between high CDAI and reduced risk of aging is particularly significant in young and middle-aged individuals and non-smokers. Therefore, consuming foods rich in CDAI components may potentially lower the risk of aging. In the future, larger-scale randomized controlled trials and further exploration in basic experiments are necessary to investigate its effectiveness and underlying mechanisms more comprehensively.

## Figures and Tables

**Figure 1 healthcare-11-02722-f001:**
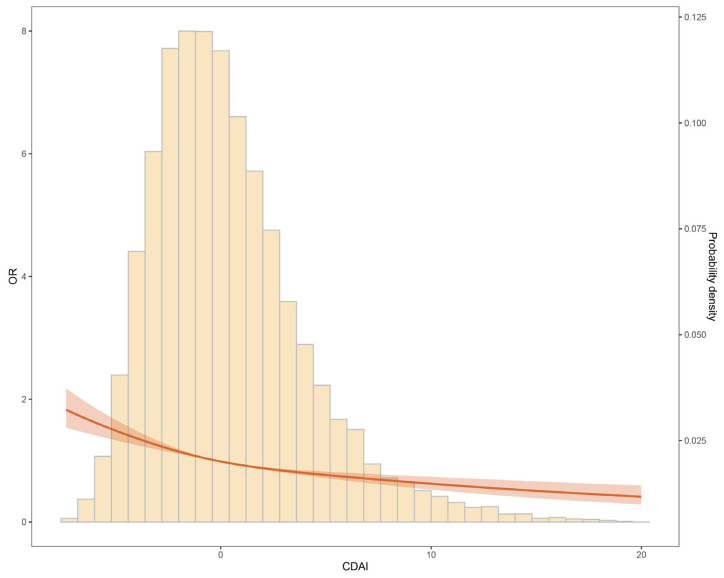
Association between CDAI and aging.

**Figure 2 healthcare-11-02722-f002:**
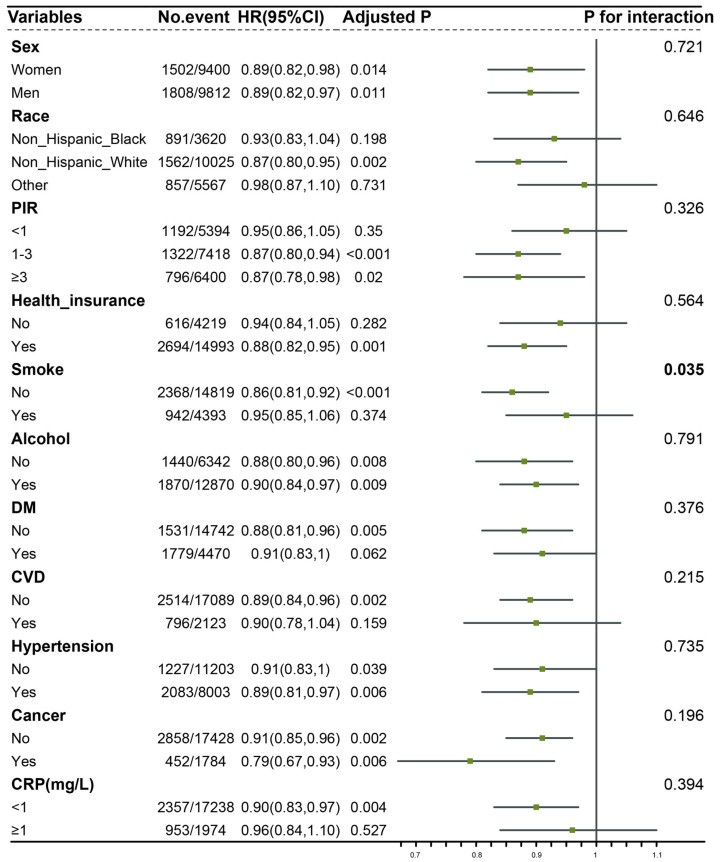
Subgroup analysis and interaction analysis. Figure legend: Model was adjusted for sex, race, BMI, education level, PIR, health insurance, smoke, alcohol, DM, CVD, hypertension, cancer history, CRP, and albumin. Abbreviation: PIR: poverty income ratio, DM: diabetes mellitus, CVD: cardiovascular disease, CRP: C-reactive protein.

**Figure 3 healthcare-11-02722-f003:**
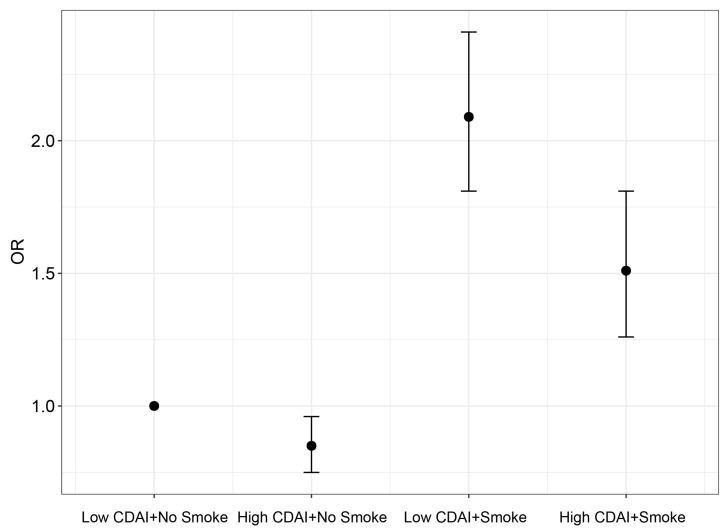
Joint analysis. Figure legend: Model was adjusted for sex, race, BMI, education level, PIR, health insurance, smoke, alcohol, DM, CVD, hypertension, cancer history, CRP, and albumin.

**Table 1 healthcare-11-02722-t001:** Baseline characteristics.

Characteristic	Level	CDAI					
		Q1	Q2	Q3	Q4	Q5	*p*
N		3842 (28,879,833)	3843 (31,351,729)	3841 (33,353,289)	3843 (35,393,618)	3843 (36,306,973)	
Age (Year)		47.25 (17.03)	47.21 (16.81)	47.06 (16.01)	46.34 (15.91)	44.89 (15.32)	<0.001
Phenoage (Year)		42.19 (19.92)	41.66 (19.37)	40.87 (18.30)	39.93 (18.04)	37.92 (17.35)	<0.001
Sex	Men	1759 (41.8)	1952 (47.9)	2046 (51.9)	1989 (50.6)	2066 (54.5)	<0.001
Race	Non-Hispanic Black	928 (14.2)	710 (10.1)	670 (9.3)	658 (8.7)	654 (8.5)	<0.001
	Non-Hispanic White	1752 (67.5)	1970 (72.3)	2059 (74.8)	2104 (74.8)	2140 (76.2)	
	Other	1162 (18.3)	1163 (17.5)	1112 (16.0)	1081 (16.5)	1049 (15.3)	
BMI (kg/m^2^)		28.56 (6.55)	28.62 (6.59)	28.62 (6.48)	28.48 (6.28)	28.22 (6.73)	0.083
Waist circumference (cm)		97.37 (15.65)	98.02 (15.98)	98.11 (15.59)	97.68 (15.82)	96.88 (16.36)	0.015
Education level	High school and above	1400 (45.2)	1721 (53.4)	1947 (59.8)	2046 (61.3)	2194 (65.7)	<0.001
PIR	<1	1433 (27.3)	1142 (20.2)	998 (17.1)	943 (16.4)	878 (15.1)	<0.001
	1-2.99	1536 (39.2)	1539 (38.4)	1502 (35.3)	1429 (34.7)	1412 (33.1)	
	≥3	873 (33.5)	1162 (41.4)	1341 (47.5)	1471 (48.9)	1553 (51.8)	
Health insurance	Yes	2876 (78.1)	2994 (81.5)	3055 (83.4)	3051 (83.2)	3017 (82.7)	<0.001
Smoke	Yes	1124 (32.5)	899 (24.7)	851 (23.4)	754 (19.8)	765 (19.1)	<0.001
Alcohol use	Yes	2276 (65.5)	2478 (69.6)	2626 (73.8)	2684 (73.8)	2806 (77.2)	<0.001
DM	Yes	1035 (20.0)	967 (19.7)	900 (17.9)	826 (16.9)	742 (15.8)	<0.001
CVD	Yes	572 (11.5)	496 (10.2)	417 (8.0)	353 (7.2)	285 (5.4)	<0.001
Hypertension	Yes	1800 (40.6)	1643 (36.5)	1637 (37.0)	1531 (34.5)	1398 (32.6)	<0.001
Cancer history	Yes	355 (9.8)	376 (9.5)	396 (9.4)	339 (8.3)	318 (7.4)	0.012
C reactive protein(mg/L)		0.48 (0.88)	0.44 (0.85)	0.39 (0.77)	0.37 (0.68)	0.35 (0.74)	<0.001
Albumin(g/dL)		42.45 (3.25)	42.47 (3.19)	42.76 (3.10)	42.85 (3.12)	43.11 (3.13)	<0.001

Abbreviation: CDAI: composite dietary antioxidant index, PIR: poverty income ratio, DM: diabetes mellitus, CVD: cardiovascular disease.

**Table 2 healthcare-11-02722-t002:** Association of composite dietary antioxidant index and aging.

CDAI	Model 1		Model 2		Model 3	
	OR (95% CI)	*p*	OR (95% CI)	*p*	OR (95% CI)	*p*
Per SD	0.81 (0.77, 0.87)	<0.001	0.89 (0.84, 0.95)	0.001	0.90 (0.84, 0.96)	0.001
Cutoff						
Low	Ref.		Ref.		Ref.	
High	0.69 (0.62, 0.78)	<0.001	0.81 (0.71, 0.91)	0.001	0.82 (0.72, 0.93)	0.003
Quantiles						
Q1	Ref.		Ref.		Ref.	
Q2	0.83 (0.72, 0.94)	0.005	0.90 (0.77, 1.04)	0.146	0.89 (0.76, 1.04)	0.141
Q3	0.73 (0.61, 0.86)	<0.001	0.84 (0.70, 1.01)	0.061	0.87 (0.70, 1.06)	0.163
Q4	0.66 (0.56, 0.79)	<0.001	0.80 (0.67, 0.96)	0.016	0.82 (0.69, 0.98)	0.032
Q5	0.52 (0.44, 0.60)	<0.001	0.66 (0.56, 0.78)	<0.001	0.68 (0.57, 0.81)	<0.001
P for trend		<0.001		<0.001		0.014

Model 1 was a crude model. Model 2 was adjusted for sex, race, BMI, education level, PIR, health insurance, smoke, and alcohol. Model 3 was adjusted for model 2 and DM, CVD, hypertension, cancer history, CRP, and albumin.

**Table 3 healthcare-11-02722-t003:** Association of composite dietary antioxidant index and aging stratified by age.

CDAI	Model 1		Model 2		Model 3	
	OR (95% CI)	*p*	OR (95% CI)	*p*	OR (95% CI)	*p*
<45 years						
Per SD	0.83 (0.75, 0.91)	<0.001	0.90 (0.81, 1.00)	0.045	0.89 (0.80, 0.99)	0.040
Cutoff						
Low	Ref.		Ref.		Ref.	
High	0.67 (0.56, 0.80)	<0.001	0.76 (0.62, 0.92)	0.005	0.73 (0.60, 0.89)	0.002
Quantiles						
Q1	Ref.		Ref.		Ref.	
Q2	0.79 (0.61, 1.00)	0.053	0.87 (0.66, 1.15)	0.323	0.89 (0.67, 1.17)	0.393
Q3	0.77 (0.60, 1.00)	0.048	0.89 (0.67, 1.17)	0.385	0.88 (0.66, 1.16)	0.354
Q4	0.63 (0.47, 0.83)	0.002	0.78 (0.57, 1.07)	0.120	0.80 (0.58, 1.11)	0.173
Q5	0.54 (0.42, 0.70)	<0.001	0.68 (0.51, 0.91)	0.010	0.65 (0.48, 0.89)	0.007
45–59 years						
Per SD	0.84 (0.75, 0.94)	0.004	0.89 (0.8, 1.00)	0.046	0.90 (0.80, 1.00)	0.052
Cutoff	Ref.		Ref.		Ref.	
Low						
High	0.76 (0.61, 0.94)	0.012	0.84 (0.66, 1.06)	0.144	0.87 (0.67, 1.12)	0.280
Quantiles						
Q1	Ref.		Ref.		Ref.	
Q2	0.85 (0.61, 1.17)	0.316	0.98 (0.68, 1.40)	0.891	0.99 (0.68, 1.44)	0.957
Q3	0.70 (0.51, 0.97)	0.033	0.86 (0.61, 1.22)	0.394	0.92 (0.63, 1.34)	0.645
Q4	0.70 (0.51, 0.96)	0.028	0.84 (0.60, 1.18)	0.316	0.87 (0.62, 1.23)	0.423
Q5	0.55 (0.42, 0.73)	<0.001	0.68 (0.50, 0.92)	0.013	0.69 (0.48, 0.98)	0.041
60–74 years						
Per SD	0.79 (0.69, 0.91)	0.001	0.89 (0.79, 1.01)	0.069	0.91 (0.8, 1.05)	0.192
Cutoff						
Low	Ref.		Ref.		Ref.	
High	0.66 (0.54, 0.80)	<0.001	0.82 (0.67, 1.00)	0.051	0.84 (0.68, 1.04)	0.104
Quantiles						
Q1	Ref.		Ref.		Ref.	
Q2	0.87 (0.67, 1.14)	0.324	0.93 (0.69, 1.25)	0.621	0.87 (0.64, 1.19)	0.389
Q3	0.71 (0.56, 0.90)	0.005	0.83 (0.64, 1.06)	0.137	0.80 (0.60, 1.07)	0.131
Q4	0.64 (0.48, 0.85)	0.002	0.83 (0.62, 1.10)	0.192	0.83 (0.62, 1.12)	0.217
Q5	0.47 (0.33, 0.67)	<0.001	0.61 (0.42, 0.89)	0.011	0.64 (0.43, 0.95)	0.029
75–84 years						
Per SD	0.92 (0.81, 1.05)	0.234	0.94 (0.82, 1.08)	0.367	0.95 (0.83, 1.10)	0.480
Cutoff						
Low	Ref.		Ref.		Ref.	
High	0.98 (0.77, 1.24)	0.837	1.02 (0.79, 1.31)	0.877	1 (0.78, 1.28)	0.999
Quantiles						
Q1	Ref.		Ref.		Ref.	
Q2	0.82 (0.57, 1.2)	0.302	0.74 (0.51, 1.08)	0.116	0.71 (0.48, 1.03)	0.067
Q3	0.85 (0.60, 1.19)	0.330	0.8 (0.56, 1.15)	0.218	0.80 (0.54, 1.16)	0.234
Q4	0.91 (0.65, 1.27)	0.573	0.9 (0.64, 1.27)	0.557	0.88 (0.63, 1.23)	0.455
Q5	0.84 (0.53, 1.33)	0.453	0.84 (0.54, 1.32)	0.440	0.85 (0.52, 1.38)	0.495

Model 1 was a crude model. Model 2 was adjusted for sex, race, BMI, education level, PIR, health insurance, smoke, and alcohol. Model 3 was adjusted for model 2 and DM, CVD, hypertension, cancer history, CRP, and albumin.

## Data Availability

The datasets used and/or analyzed during the current study are available from the corresponding author on reasonable request.

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
