# Peer review of "Relationship between Composite Dietary Antioxidant Index and Aging"

_healthcare, 2023, doi:10.3390/healthcare11202722_

Round 1
Reviewer 1 Report
In this study, authors analyze the association of between the composite dietary antioxidant index (CDAI) and aging. Although the work is relevant, there are some issues that need to be addressed before it can be accepted for publication:
1. Manuscript requires proofreading for English language.
2. Presentation of data should be improved for legibility.
3. Authors mentioned that a strict inclusion criteria was applied while selecting the participants however authors never describe the meaning of inclusion criteria. It is important to mention that in the manuscript for proper evaluation of results.
English language needs to be improved.
Author Response
Dear editor and reviewers,
Warm greetings! We are very grateful to the editors and reviewers for their comments on this manuscript, which we think are very important and useful. We have made detailed explanations and amendments (marked in red) to these comments. We hope that these revisions can make the editors and reviewers more satisfied and make the manuscript better. If there are any other comments, please give us a opportunity to revise again. I would like to express my sincere thanks for your work and comments.
Reply to Review#1
In this study, authors analyze the association of between the composite dietary antioxidant index (CDAI) and aging. Although the work is relevant, there are some issues that need to be addressed before it can be accepted for publication:
- Manuscript requires proofreading for English language.
Response: Thank you for your language-related suggestions. We are not native English speakers, so we enlisted the help of native English speakers to make the necessary revisions (Mainly in terms of grammar) to our manuscript. We hope these changes will make the entire manuscript more readable.
- Presentation of data should be improved for legibility.
Response: Thank you for your comment. I apologize for not clearly presenting the table/figure captions. Therefore, based on your comment, we have added corresponding captions below Table 1, Figure 2, and Figure 3. We have also made adjustments to the supplementary files, where we remade Figure S1 and expanded the abbreviations in Figure S1 and provided detailed explanations for the models below Figure S3, Table S1, and Table S2.
- Authors mentioned that a strict inclusion criteria was applied while selecting the participants however authors never describe the meaning of inclusion criteria. It is important to mention that in the manuscript for proper evaluation of results.
Response: Thank you for your comment. We have added the relevant inclusion and exclusion criteria and explained the reasons and meaning for considering these factors in the supplementary methods.
|
Original |
Revised |
|
After applying strict inclusion and exclusion criteria, a total of 19,212 participants were included in this study (Figure S1). |
1.Exclusion criteria included: (1) Participants lacking covariates required for the calculation of phenotypic age (albumin, creatinine, glucose, C-reactive protein [CRP], lymphocyte percentage, mean cell volume, red cell distribution width, alkaline phosphatase, and white blood cell count); (2) Participants lacking covariates for model adjustments (factors such as smoking, alcohol consumption, comorbidity information, health insurance, and income, Supplementary method 1). After applying strict inclusion and exclusion criteria, a total of 19,212 participants were included in this study (Figure S1). 2.Supplementary method 1 These factors have been widely established to be associated with aging. David et al. demonstrated that smoking reduces average lifespan by 7 years, and tobacco consumption shortens healthy life expectancy by 14 years [1]. Polliana et al. found a close association between alcohol consumption and brain aging and cognitive impairments[2]. Furthermore, comorbidities such as hypertension, diabetes, and coronary heart disease are considered accompanying symptoms of aging. Regarding social characteristics, income, health insurance, and educational level have also been linked to individual aging[3-4]. [1] Bernhard D, Moser C, Backovic A, Wick G. Cigarette smoke--an aging accelerator? Exp Gerontol. 2007 Mar;42(3):160-5. doi: 10.1016/j.exger.2006.09.016. [2] Nunes PT, Kipp BT, Reitz NL, Savage LM. Aging with alcohol-related brain damage: Critical brain circuits associated with cognitive dysfunction. Int Rev Neurobiol. 2019;148:101-168. doi: 10.1016/bs.irn.2019.09.002. [3] Pothisiri W, Prasitsiriphon O, Aekplakorn W. Extent of aging across education and income subgroups in Thailand: Application of a characteristic-based age approach. PLoS One. 2020 Dec 8;15(12):e0243081. doi: 10.1371/journal.pone.0243081. [4] Parker SW, Saenz J, Wong R. Health Insurance and the Aging: Evidence From the Seguro Popular Program in Mexico. Demography. 2018 Feb;55(1):361-386. doi: 10.1007/s13524-017-0645-4. |
Best wishes,
Haiting Wang.
Reviewer 2 Report
The study highlights significant findings but following clarifications are required:
1. “Participants were divided into five groups (Q1 146 to Q5) based on quintiles: Q1 (-7.35 to -2.63), Q2 (-2.63 to -1.01), Q3 (-1.01 to 0.65), Q4 (0.65 147 to 3.06), and Q5 (3.06 to 36.70).” What are these Quintiles? How are they calculated? It is not mentioned in methods
2. Line 153 “Association of composite dietary antioxidant index and aging” seems to be like a subheading
3. What are the inclusion and exclusion criteria for “data from NHANES between 2001 and 2010”? Its not clear.
4. Figure legends are not clear
needs some minor improvements
Author Response
Dear editor and reviewers,
Warm greetings! We are very grateful to the editors and reviewers for their comments on this manuscript, which we think are very important and useful. We have made detailed explanations and amendments (marked in red) to these comments. We hope that these revisions can make the editors and reviewers more satisfied and make the manuscript better. If there are any other comments, please give us a opportunity to revise again. I would like to express my sincere thanks for your work and comments.
Reply to Review#2
The study highlights significant findings but following clarifications are required:
1.“Participants were divided into five groups (Q1 to Q5) based on quintiles: Q1 (-7.35 to -2.63), Q2 (-2.63 to -1.01), Q3 (-1.01 to 0.65), Q4 (0.65 to 3.06), and Q5 (3.06 to 36.70).” What are these Quintiles? How are they calculated? It is not mentioned in methods.
Response: Thank you for your comments. I apologize for not clearly describing the source of the quintiles in the methods and results sections. Here, these quintiles are based on the CDAI quintiles. In order to more elaborately describe the relationship between CDAI and aging, we divided each participant's CDAI into quintiles, ranging from the lowest to the highest. We have made corresponding revisions in both the methods and results sections based on your suggestions.
|
Original |
Revised |
|
1.Weighted logistic regression analysis was employed to describe the association between CDAI or its components and aging, presented as odds ratio (OR) with corresponding 95% confidence intervals (CI). 2. Participants were divided into five groups (Q1 to Q5) based on quintiles… |
1.Weighted logistic regression analysis was employed to describe the association between CDAI or its components and aging, presented as odds ratio (OR) with corresponding 95% confidence intervals (CI). In the logistic regression analysis, we examined the relationship between CDAI as a continuous variable, as a binary variable (based on the median), and as a categorical variable with five groups (based on quintiles) and aging. 2. Participants were divided into five groups (Q1 to Q5) based on quintiles of CDAI…
|
2.Line 153 “Association of composite dietary antioxidant index and aging” seems to be like a subheading
Response: Thank you for your comment. This sentence does indeed serve as a subheading, and we have made the corresponding revisions.
3.What are the inclusion and exclusion criteria for “data from NHANES between 2001 and 2010”? Its not clear.
Response: Thank you for your comment. As mentioned in the Methods section of the manuscript, NHANES collects participant data in cycles spanning two years each. The data used in this study includes five survey cycles from 2001 to 2010, specifically 2001-2002, 2003-2004, 2005-2006, 2007-2008, and 2009-2010. The reason for including data from these years is twofold: (1) After 2010, there were changes in the measurement methods for CRP, including the measurement equipment and technique, and the shift to high-sensitivity CRP. (2) The construction of the phenotypic age was based on data from 2001 to 2010, ensuring the universality and accuracy of the phenotypic age calculation equation [1-2]. Therefore, we selected data only from the years 2001 to 2010 to maintain consistency and reliability in our study.
- Levine ME, Lu AT, Quach A et al. An epigenetic biomarker of aging for lifespan and healthspan. Aging (Albany NY) 2018; 10: 573-591.
- Liu Z, Kuo PL, Horvath S et al. A new aging measure captures morbidity and mortality risk across diverse subpopulations from NHANES IV: A cohort study. PLoS Med 2018; 15: e1002718.
4.Figure legends are not clear
Response: Thank you for your comment. I apologize for not clearly presenting the table/figure captions. We have made the necessary revisions here.
|
Original |
Revised |
|
None |
Figure 2 legend: Model was adjusted for sex, race, BMI, education level, PIR, health insurance, smoke, alcohol, DM, CVD, hypertension, cancer history, CRP and albumin. Abbreviation: PIR: poverty income ratio,DM: diabetes mellitus, CVD: cardiovascular disease, CRP: C-reactive protein. Figure3 legend: Model was adjusted for sex, race, BMI, education level, PIR, health insurance, smoke, alcohol, DM, CVD, hypertension, cancer history, CRP and albumin. |
Best wishes,
Haiting Wang.
Reviewer 3 Report
The aim of this study was to investigate the potential association between CDAI and aging using the National Health and Nutrition Examination Survey (NHANES) database to provide reliable dietary guidance for anti-aging. This article is interesting and well organized, but has the following issues for improvement:
1、Header " Healthcare 2021, 9, x. https", should the year be 2023?
2、Font inconsistency in Abstract;
3、In Figure S1, should CVD be given its full name and is this covariate also included in the study?
4、Please explain what data is represented by Ref. in Table S2?
5、Line 144, please explain the meaning of "weighted: 165,285,442 individuals";
6、Line 146-147,What are the indicators that participants are categorised into 5 groups? Is it CDAI value or age?
7、Is Line 153 a title or a body?
8、Please explain what the data such as 3842 (28879833), 3843 (31351729), etc. for Q1-Q5 in row N of Table 1 represent? Also for the Race row is there only one p-value?
9、Line 159-168, this result is for model 3 and should be clarified, as well as the authors' methodology for building the 3 models should be explained in the methodology;
10、Line 169-176, the authors should state which model this result is for? For Figure S3 and Table S1 it should also be made clear which of the 3 models it is for.
Author Response
Dear editor and reviewers,
Warm greetings! We are very grateful to the editors and reviewers for their comments on this manuscript, which we think are very important and useful. We have made detailed explanations and amendments (marked in red) to these comments. We hope that these revisions can make the editors and reviewers more satisfied and make the manuscript better. If there are any other comments, please give us a opportunity to revise again. I would like to express my sincere thanks for your work and comments.
Reply to Review#3
The aim of this study was to investigate the potential association between CDAI and aging using the National Health and Nutrition Examination Survey (NHANES) database to provide reliable dietary guidance for anti-aging. This article is interesting and well organized, but has the following issues for improvement:
- Header " Healthcare 2021, 9, x. https", should the year be 2023?
Response: Thank you for your comment. This is the template for the journal. In my reply to the editorial department, I will remind them to make the necessary modifications.
- Font inconsistency in Abstract;
Response: Thank you for your comment. I have standardized the font in the abstract.
- In Figure S1, should CVD be given its full name and is this covariate also included in the study?
Response: Thank you for your comment. We have provided the full names for these abbreviations. Additionally, all of these variables were included as covariates in the logistic regression calculations. These factors have been widely established to be associated with aging. David et al. demonstrated that smoking reduces average lifespan by 7 years, and tobacco consumption shortens healthy life expectancy by 14 years [1]. Polliana et al. found a close association between alcohol consumption and brain aging and cognitive impairments [2]. Furthermore, comorbidities such as hypertension, diabetes, and coronary heart disease are considered accompanying symptoms of aging. Regarding social characteristics, income, health insurance, and educational level have also been linked to individual aging [3-4].
[1] Bernhard D, Moser C, Backovic A, Wick G. Cigarette smoke--an aging accelerator? Exp Gerontol. 2007 Mar;42(3):160-5. doi: 10.1016/j.exger.2006.09.016.
[2] Nunes PT, Kipp BT, Reitz NL, Savage LM. Aging with alcohol-related brain damage: Critical brain circuits associated with cognitive dysfunction. Int Rev Neurobiol. 2019;148:101-168. doi: 10.1016/bs.irn.2019.09.002.
[3] Pothisiri W, Prasitsiriphon O, Aekplakorn W. Extent of aging across education and income subgroups in Thailand: Application of a characteristic-based age approach. PLoS One. 2020 Dec 8;15(12):e0243081. doi: 10.1371/journal.pone.0243081.
[4] Parker SW, Saenz J, Wong R. Health Insurance and the Aging: Evidence From the Seguro Popular Program in Mexico. Demography. 2018 Feb;55(1):361-386. doi: 10.1007/s13524-017-0645-4.
4、Please explain what data is represented by Ref. in Table S2?
Response: Thank you for your comment. I'm sorry for any confusion caused by the abbreviation. "Ref." refers to the reference group, which consists of patients with a low CDAI or Q1 CDAI. For example, in the column "Excluding tumor patients," compared to the low CDAI group (Reference), the odds ratio for the increased aging risk in the high CDAI group is 0.81 (0.72, 0.92). We have now updated all instances of "Ref." to "Reference" for clarity.
5、Line 144, please explain the meaning of "weighted: 165,285,442 individuals";
Response: Thank you for your comment. As mentioned in the methods section, NHANES is a study cohort conducted through a stratified survey, which means that the weighting in NHANES is a method used to adjust the sample data to reflect the entire U.S. population. The purpose of this weighting process is to ensure that the study results statistically represent the entire U.S. population, not just the sample population participating in the survey. NHANES employs a complex multi-stage sampling design to select participants, which means that people from different groups have different probabilities of being selected. To conduct effective statistical analysis, it is necessary to weight these sampling probabilities to more accurately estimate specific parameters in the U.S. population and predict overall trends. By applying these weights, NHANES allows researchers to better estimate the health and nutritional status of the U.S. population and various health issues related to it. This helps ensure that survey results are more representative and applicable.
According to the NHANES recommendations, each participant here is assigned an appropriate weight and is used in calculations. Therefore, the 19,212 participants in this analysis represent a weighted sample of 165,285,442 individuals from various regions and strata in the United States.
- Line 146-147,What are the indicators that participants are categorised into 5 groups? Is it CDAI value or age?
Response: Thank you for your comment. I'm sorry I didn't clarify this issue in detail. Low CDAI and high CDAI are distinguished based on the median, and all five quartiles (Q1-5) are divided according to CDAI. We have provided additional details in the methods section.
|
Original |
Revised |
|
Weighted logistic regression analysis was employed to describe the association between CDAI or its components and aging, presented as odds ratio (OR) with corresponding 95% confidence intervals (CI).
|
Weighted logistic regression analysis was employed to describe the association between CDAI or its components and aging, presented as odds ratio (OR) with corresponding 95% confidence intervals (CI). In the logistic regression analysis, we examined the relationship between CDAI as a continuous variable, as a binary variable (based on the median), and as a categorical variable with five groups (based on quintiles) and aging.
|
- Is Line 153 a title or a body?
Response: Thank you for your comment. This is a subheading that was overlooked during the formatting revisions at the journal.
- Please explain what the data such as 3842 (28879833), 3843 (31351729), etc. for Q1-Q5 in row N of Table 1 represent? Also for the Race row is there only one p-value?
Response: Thank you for your comment.As mentioned in comment 5, this is a stratified survey research cohort, which means that the weighting in NHANES is a method used to adjust sample data to reflect the entire U.S. population. Therefore, the numbers shown in parentheses represent the actual population that these samples are meant to represent. In the racial category, we grouped individuals from the five groups together for the chi-squared test. We believe that this approach is statistically valid and can reflect the racial differences among the five CDAI groups, as demonstrated in previous studies [1].
[1]Liu CA, Liu T, Ge YZ, et al. Muscle distribution in relation to all-cause and cause-specific mortality in young and middle-aged adults. J Transl Med. 2023 Feb 25;21(1):154. doi: 10.1186/s12967-023-04008-7
9、Line 159-168, this result is for model 3 and should be clarified, as well as the authors' methodology for building the 3 models should be explained in the methodology;
Response: Thank you for your reminder. Clarifying our model is indeed crucial. Following your advice, we have added relevant content in the methods and results sections.
|
Original |
Revised |
|
1.Weighted logistic regression analysis was employed to describe the association between CDAI (as continue, by median, or by quintile) or its components and aging, presented as odds ratio (OR) with corresponding 95% confidence intervals (CI). 2.When treated as a continuous variable and adjusted for confounding factors, each standard deviation increase... |
1.Weighted logistic regression analysis was employed to describe the association between CDAI (as continue, by median, or by quintile) or its components and aging, presented as odds ratio (OR) with corresponding 95% confidence intervals (CI). We constructed three models to comprehensively evaluate the relationship between them. Modle 1 was crude model. Model 2 was adjusted for sex, race, BMI, education level, PIR, Health insurance, Smoke, and alcohol. Model 3 was adjusted for model 2 and DM, CVD,hypertension, cancer history, CRP and albumin. 2.When treated as a continuous variable and adjusted for confounding factors (sex, race, BMI, education level, PIR, health insurance, smoke, alcohol, DM, CVD, hypertension, cancer history, CRP and albumin.), each standard deviation increase... |
10、Line 169-176, the authors should state which model this result is for? For Figure S3 and Table S1 it should also be made clear which of the 3 models it is for.
Response: Thank you for your reminder. I'm sorry that I didn't label these contents in the figure and table legends. Now, I have added legends for all the models in the supplementary material (Figure S3, Table S1, 2).
|
Original |
Revised |
|
None |
Notes: Model was adjusted for sex, race, BMI, education level, PIR, health insurance, smoke, alcohol, DM, CVD, hypertension, cancer history, CRP and albumin. |
Best wishes,
Haiting Wang.
Reviewer 4 Report
The article describes the research carried out to investigate the association between the Composite Dietary Antioxidant Index (CDAI) and ageing using data from five cycles of the National Health and Nutrition Examination Survey (NHANES), specifically from 2001 to 2010, a survey designed to investigate the health and nutrition of the US population. The data used correspond to a total of 19 212 participants, a sufficient sample for the cross-sectional study presented in the paper. Dietary data obtained from participants' 24-hour dietary reminders, collected on two occasions with an interval of 3-10 days, were considered. CDAI was calculated using a validated formula. Likewise, phenotypic age was calculated using a formula also validated and collected in the literature, and the degree of ageing was determined through what the authors call "phenotypic age acceleration" by subtracting chronological age from phenotypic age, thus defining phenotypic ageing when the value obtained is greater than 0 and phenotypic rejuvenation if this value is less than or equal to 0.
The work is particularly interesting both for the subject matter dealt with and for the methodological procedure followed, as well as for the repercussions it could have on ageing, a highly topical issue. Importance and interest that the authors adequately justify in the introduction. Likewise, both the objective of the study and the research questions are clearly stated. Therefore, the interest of the study is fully justified and sufficiently argued.
Regarding the methodological development, the statistical analyses performed are described in detail (Pearson correlation analysis to examine the linear relationship between CDAI and phenotypic age, restrictive cubic spline regression to study the relationship between CDAI or its components and ageing, weighted logistic regression analysis to examine the relationship between CDAI or its components and ageing, weighted logistic regression analysis to examine the relationship between CDAI and phenotypic ageing, and weighted logistic regression analysis to examine the relationship between CDAI and ageing); weighted logistic regression analyses to describe the association between CDAI or its components and ageing; and interaction analyses to explore potential modifiers in these relationships, including age, sex, race, PIR, health insurance, smoking, alcohol consumption, DM, hypertension, cancer and CRP (with a cut-off point of 1 mg/L). Subgroup analyses were also performed according to these factors to describe the relationship between CDAI and ageing in different strata and, in addition, pooled analyses were performed to elucidate the role of modifiers in the presence of interaction effects. Finally, the authors have conducted sensitivity analyses to assess the robustness of the results. All these statistical treatments give sufficient robustness to the study and, therefore, to its results, which are well presented in the tables and graphs contained in the article and in the supplementary material.
The discussion is adequate on the basis of the results obtained in the research. As for the limitations, the authors are aware of the main limitations of a study of these characteristics and this is reflected in the work: not being able to establish causal relationships as occurs in all cross-sectional studies, not considering the consumption of antioxidants in the participants as well as only considering dietary data from a single point in time. As they rightly point out, more long-term dietary research is needed to provide more accurate assessments.
Finally, I also consider the conclusions drawn from the study to be appropriate.
Based on the above, it is recommended: Accept in present form.
Author Response
Dear editor and reviewers,
Warm greetings! We are very grateful to the editors and reviewers for their comments on this manuscript, which we think are very important and useful. We have made detailed explanations and amendments (marked in red) to these comments. We hope that these revisions can make the editors and reviewers more satisfied and make the manuscript better. If there are any other comments, please give us a opportunity to revise again. I would like to express my sincere thanks for your work and comments.
Reply to Review#4
The article describes the research carried out to investigate the association between the Composite Dietary Antioxidant Index (CDAI) and ageing using data from five cycles of the National Health and Nutrition Examination Survey (NHANES), specifically from 2001 to 2010, a survey designed to investigate the health and nutrition of the US population. The data used correspond to a total of 19 212 participants, a sufficient sample for the cross-sectional study presented in the paper. Dietary data obtained from participants' 24-hour dietary reminders, collected on two occasions with an interval of 3-10 days, were considered. CDAI was calculated using a validated formula. Likewise, phenotypic age was calculated using a formula also validated and collected in the literature, and the degree of ageing was determined through what the authors call "phenotypic age acceleration" by subtracting chronological age from phenotypic age, thus defining phenotypic ageing when the value obtained is greater than 0 and phenotypic rejuvenation if this value is less than or equal to 0.
The work is particularly interesting both for the subject matter dealt with and for the methodological procedure followed, as well as for the repercussions it could have on ageing, a highly topical issue. Importance and interest that the authors adequately justify in the introduction. Likewise, both the objective of the study and the research questions are clearly stated. Therefore, the interest of the study is fully justified and sufficiently argued.
Regarding the methodological development, the statistical analyses performed are described in detail (Pearson correlation analysis to examine the linear relationship between CDAI and phenotypic age, restrictive cubic spline regression to study the relationship between CDAI or its components and ageing, weighted logistic regression analysis to examine the relationship between CDAI or its components and ageing, weighted logistic regression analysis to examine the relationship between CDAI and phenotypic ageing, and weighted logistic regression analysis to examine the relationship between CDAI and ageing); weighted logistic regression analyses to describe the association between CDAI or its components and ageing; and interaction analyses to explore potential modifiers in these relationships, including age, sex, race, PIR, health insurance, smoking, alcohol consumption, DM, hypertension, cancer and CRP (with a cut-off point of 1 mg/L). Subgroup analyses were also performed according to these factors to describe the relationship between CDAI and ageing in different strata and, in addition, pooled analyses were performed to elucidate the role of modifiers in the presence of interaction effects. Finally, the authors have conducted sensitivity analyses to assess the robustness of the results. All these statistical treatments give sufficient robustness to the study and, therefore, to its results, which are well presented in the tables and graphs contained in the article and in the supplementary material.
The discussion is adequate on the basis of the results obtained in the research. As for the limitations, the authors are aware of the main limitations of a study of these characteristics and this is reflected in the work: not being able to establish causal relationships as occurs in all cross-sectional studies, not considering the consumption of antioxidants in the participants as well as only considering dietary data from a single point in time. As they rightly point out, more long-term dietary research is needed to provide more accurate assessments.
Finally, I also consider the conclusions drawn from the study to be appropriate.
Based on the above, it is recommended: Accept in present form.
Response: Thank you very much for your recognition of this study. Our research emphasizes that the association between high CDAI and reduced risk of aging, which is particularly significant in young and middle-aged individuals and non-smokers. Therefore, consuming foods rich in CDAI components may potentially lower the risk of aging.
Best wishes,
Haiting Wang.
Round 2
Reviewer 3 Report
The author has followed my suggestions to improve the article and i personally believe that the article can now be accepted.